Methods

# Human iPS cell–derived respiratory organoids as a model for respiratory syncytial virus infection

Rina Hashimoto[1,2],†, Yukio Watanabe[1],†, Abeer Keshta[1] , Masaya Sugiyama[3], Yuki Kitai[4], Ai Hirabayashi[5,6], Naoko Yasuhara[1], Shiho Morimoto[1], Ayaka Sakamoto[1], Yasufumi Matsumura[7] , Hidekazu Nishimura[8], Takeshi Noda[5,6,9], Takuya Yamamoto[1,10,11] , Miki Nagao[7] , Makoto Takeda[4], Kazuo Takayama[1,2]

**Respiratory syncytial virus (RSV) is a seasonal respiratory pathogen that primarily affects young children, potentially causing severe lower respiratory tract disease. Despite the high disease burden, understanding of RSV pathophysiology remains limited. To address this, advanced RSV infection models are needed. Whereas HEp-2 cells are widely used because of their high susceptibility to RSV, they do not accurately reflect the host response of the human respiratory tract. In this study, we evaluated human-induced pluripotent stem cell-derived respiratory organoids, which contain respiratory epithelial cells, immune cells, fibroblasts, and vascular endothelial cells, for their potential to model RSV infection and support pharmaceutical research. RSV-infected organoids exhibited high viral genome and protein expression, epithelial layer destruction, and increased collagen accumulation. Pro-inflammatory cytokine levels in culture supernatants also increased post-infection. Furthermore, RSV infection was significantly inhibited by monoclonal antibodies (nirsevimab, palivizumab, suptavumab, or clesrovimab), although ribavirin showed limited efficacy. These findings highlight the utility of respiratory organoids for RSV research.**

## Introduction

RSV infections carry the risk of causing severe respiratory disorders. As such, there is an urgent need to develop better therapeutic and preventive drugs for not only flu and coronavirus disease 2019 (COVID-19) but also for RSV infections. To effectively advance pharmaceutical research for RSV, it is crucial to develop a model that accurately replicates the pathophysiology of RSV infection.

Although various models have been developed to study viral respiratory infections, human respiratory organoids have attracted attention in recent years, as they allow for more accurate analysis of pathophysiology and evaluations of antiviral drugs and antibodies than cell lines. HEp-2 cells have been widely used in RSV research because of their high viral replication efficiency (Jordan, 1962); however, they are not ideal for analyzing host responses. In contrast, respiratory organoids, which contain cells that constitute the lungs and airways, are better suited for precise analysis of host responses. We have recently examined anti-severe acute respiratory syndrome coronavirus 2 (SARS-CoV-2) drugs (Hashimoto et al, 2023; Tamura et al, 2024) and antibodies (Bodie et al, 2023) using respiratory organoids developed by our group. Similarly, respiratory organoids could be applicable for developing RSV therapeutic and preventive drugs. In this study, in addition to analyzing the host response to RSV infection using human respiratory organoids, we also conducted a comparative analysis of antibodies and antiviral drugs against RSV.

Aerosolized ribavirin was previously used for treating patients at highest risk of RSV. However, given the contradicting evidence on its efficacy (Hall et al, 1983; Smith et al, 1991; Long et al, 1997; Guerguerian et al, 1999), limited use, and inconvenient route of administration, there is a need for developing newer RSV therapeutic drugs. Conversely, multiple antibodies have been developed as preventive drugs. Palivizumab (Synagis) has been used as a preventive drug against RSV infections in infants since 1998, and several new antibodies against RSV, such as nirsevimab (Beyfortus), have been developed. Whereas palivizumab requires monthly dosing, nirsevimab has a longer half-life than palivizumab, remaining effective for more than five months (Mullard, 2023), and

[1]Center for iPS Cell Research and Application (CiRA), Kyoto University, Kyoto, Japan    [2]Department of Synthetic Human Body System, Medical Research Institute, Institute of Integrated Research, Institute of Science Tokyo, Tokyo, Japan    [3]Department of Viral Pathogenesis and Controls, National Center for Global Health and Medicine, Ichikawa, Japan    [4]Department of Microbiology, Graduate School of Medicine and Faculty of Medicine, The University of Tokyo, Tokyo, Japan    [5]Laboratory of Ultrastructural Virology, Institute for Life and Medical Sciences, Kyoto University, Kyoto, Japan    [6]CREST, Japan Science and Technology Agency, Saitama, Japan    [7]Department of Clinical Laboratory Medicine, Graduate School of Medicine, Kyoto University, Kyoto, Japan    [8]Virus Research Center, Clinical Research Division, Sendai Medical Center, National Hospital Organization, Sendai, Japan    [9]Laboratory of Ultrastructural Virology, Graduate School of Biostudies, Kyoto University, Kyoto, Japan    [10]Institute for the Advanced Study of Human Biology (WPI-ASHBi), Kyoto University, Kyoto, Japan    [11]Medical-risk Avoidance based on iPS Cells Team, RIKEN Center for Advanced Intelligence Project (AIP), Kyoto, Japan

Correspondence: kazuo.takayama@cira.kyoto-u.ac.jp
†Rina Hashimoto and Yukio Watanabe are co-first authors

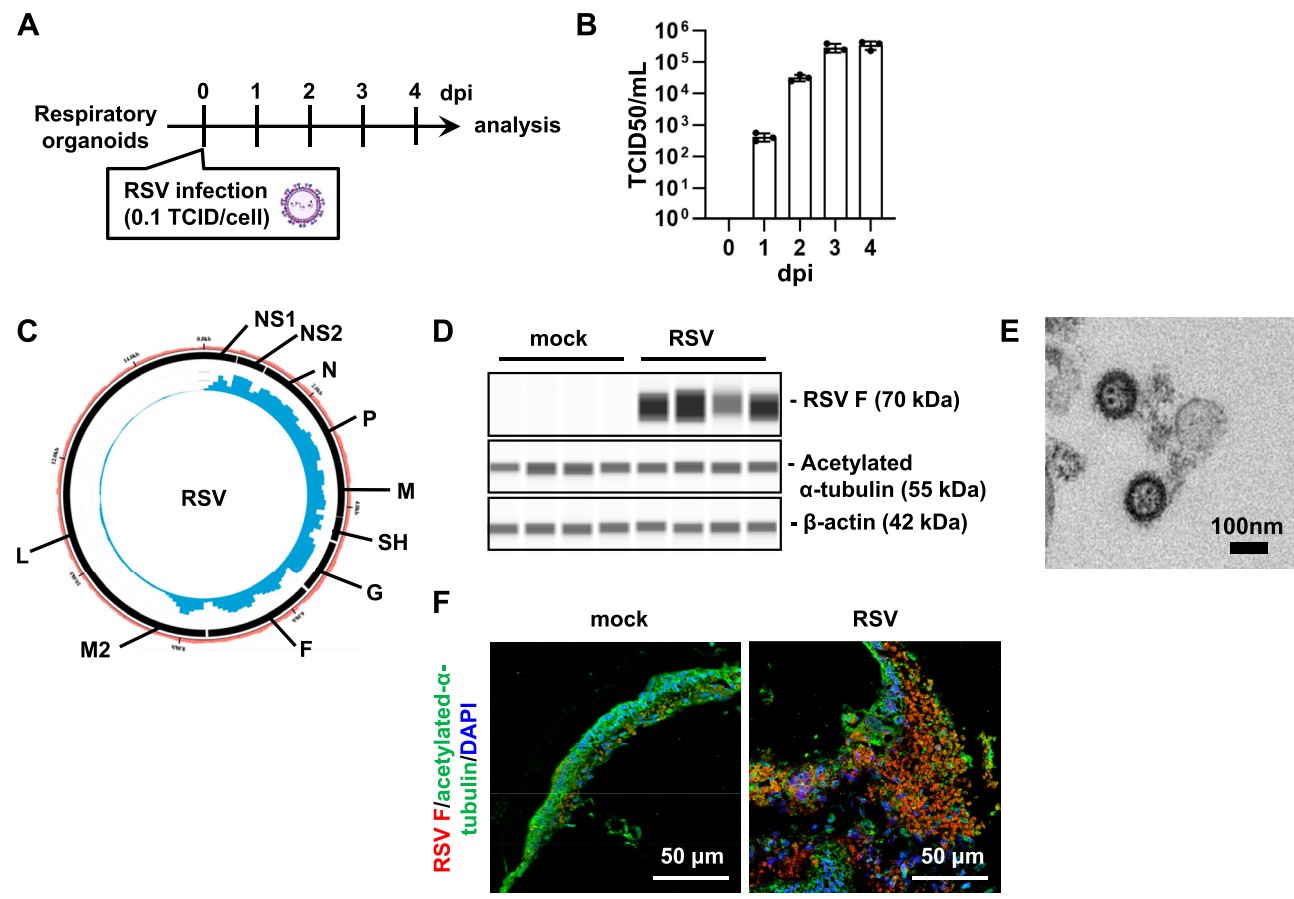

**Figure 1. RSV infects human iPS cell–derived respiratory organoids.**
**(A)** Human iPS cell–derived respiratory organoids were infected with 0.1 TCID/cell RSV (8 × 10$^4$ TCID/well) and cultured for 4 d. **(B)** TCID50 assay was temporally performed in the RSV-infected respiratory organoids. **(C)** RNA sequencing was performed for RSV-infected respiratory organoids. Circos plot showing the distribution of reads along the RSV genome. The inner blue circles indicate viral RNA-seq reads obtained from RSV-infected respiratory organoids. The outer black blocks show the coding regions of RSV. The outer pale red bar plot signifies the percentage of GC content. The Circos plot was created using Circleator (v 1.0.2) (Crabtree et al, 2014). **(D)** Expression levels of RSV F, acetylated α-tubulin, and β-actin were measured by a capillary-based immunoassay (Jess analysis). **(E)** TEM image of RSV. **(F)** Immunofluorescence analysis of acetylated α-tubulin (green) and RSV F (red) in the respiratory organoids.

is, thus, a more attractive option than palivizumab. Moving forward, to continue developing therapeutic and preventive drugs for RSV efficiently, respiratory models with high clinical predictability are necessary.

The study aimed to evaluate human-induced pluripotent stem (iPS) cell-derived respiratory organoids, which contain respiratory epithelial cells, immune cells, fibroblasts, and vascular endothelial cells, as a potential model for RSV infection and to support pharmaceutical research.

## Results

### RSV efficiently infects respiratory organoids derived from human iPS cells

We used human iPS cell–derived respiratory organoids as a model to reproduce the pathophysiology of RSV infections. Respiratory organoids contain not only respiratory epithelial cells but also contain macrophages, fibroblasts, and vascular endothelial cells, thus making it possible to reproduce pathological conditions such as inflammation. Because of the apical-out structure of the respiratory organoids used, RSV infection can be performed simply by adding the virus to the culture medium. We infected respiratory organoids with RSV-A subtype and cultured for 4 d (Fig 1A). Time-course TCID50 assays after RSV infection showed that viral titers reached a plateau exceeding 10$^5$ TCID50/ml by 3~4 dpi (Fig 1B). RNA-seq analysis was performed 4 d after RSV infection. Circos plot shows that RSV mRNA is highly expressed in respiratory organoids (Fig 1C). A capillary-based immunoassay indicated that RSV fusion (F) protein was highly expressed (Figs 1D and S1). Transmission electron microscope images also demonstrated the presence of virus particles in the infected respiratory organoids (Fig 1E). Immunofluorescence analysis revealed that RSV infection disrupted the integrity of the acetylated α-tubulin-positive cell layer (Fig 1F). These results demonstrated that the efficient RSV infection can be performed in respiratory organoids.

## Investigation of the host response to RSV infection in respiratory organoids

The host response to RSV infection was investigated in respiratory organoids. Hematoxylin and eosin and Sirius red staining showed that RSV infections disrupt the respiratory epithelial cell layer and induce collagen accumulations (Fig 2A and B). RNA-seq analysis of respiratory organoids infected with RSV revealed that 381 and 120 genes were significantly up-regulated or down-regulated, respectively (Fig 2C, Table S1). Enrichment analysis of genes up-regulated upon RSV infection revealed the top 10 gene ontology (GO) terms, implicating "innate immune response," "cellular response to IFN-γ," and "IFN-γ-mediated signaling pathway" (Figs 2D and S2A and B). In addition, when we measured the concentrations of cytokines in the culture supernatant of RSV-infected respiratory organoids, we found that the concentrations of IL-8 and IFN-γ, cytokines reported to increase during RSV infection (Zeng et al, 2011), were increased (Fig 2E). These results suggest RSV infects respiratory organoids efficiently and induces respiratory epithelial cell layer destruction, collagen accumulation, innate immune response, and inflammatory response.

## Evaluation of antibodies and antiviral drugs against RSV

We evaluated antibodies and antiviral drugs developed for RSV using respiratory organoids. Nirsevimab, suptavumab, clesrovimab, and palivizumab are antibody drugs against the RSV F protein. Palivizumab has been widely used as a preventive drug for RSV for more than 20 yr (palivizumab, a humanized respiratory syncytial virus monoclonal antibody, reduces hospitalization from respiratory syncytial virus infection in high-risk infants. The IMpact-RSV Study Group, 1998), although nirsevimab (Griffin et al, 2020, Simões et al, 2022) was recently approved by the FDA as a preventive drug for RSV. Ribavirin (Hall et al, 1983; Smith et al, 1991) is a therapeutic drug approved for use in the treatment of RSV. Using respiratory organoids, we investigated the effectiveness of antiviral antibodies and drugs against RSV (Fig 3A). The TCID50 value decreased upon treatment with all four types of antibodies, with palivizumab exhibiting the greatest reduction (Fig 3B). In contrast, ribavirin treatment resulted in minimal change in TCID50 value, suggesting that its antiviral effect is not evident in respiratory organoids (Fig 3B). Consistent with this finding, a previous study has reported that ribavirin is ineffective in RSV patients (Hall et al, 1983; Guerguerian et al, 1999). Taken together, respiratory organoids may serve as a valuable model for evaluating antibodies and antiviral drugs against RSV.

To examine the effects of anti-RSV F antibodies on host cells in respiratory organoids, single-cell RNA sequencing (scRNA-seq) analysis was conducted. Although RSV infection altered the cellular composition of respiratory organoids (Fig S3A and B and Table S2), scRNA-seq analysis was performed only once, limiting the ability to conduct statistical analysis. UMAP analysis revealed detectable levels of *RSV N* mRNA in various cell clusters (Fig S3C). scRNA-seq analysis was performed in RSV-infected respiratory organoids treated with nirsevimab or palivizumab. Because RSV has been reported to infect human respiratory epithelial cells (Zhang et al, 2002; Alcorn et al, 2005; Johnson et al, 2007; Villenave et al,

2012), we also analyzed *RSV N* expression in respiratory epithelial cells. We also confirmed that *RSV N* expression in alveolar type I and type II cells, ciliated cells, and secretory cells to be abolished by nirsevimab and palivizumab (Fig 3C). It is suggested that respiratory epithelial cells can be protected from RSV infection through nirsevimab or palivizumab treatment.

Because RSV efficiently infects respiratory epithelial cells such as alveolar type I and type II epithelial cells, ciliated cells, and secretory cells, we also analyzed host cell responses in these cells. RSV infections induced innate immune response-related genes (*ISG15* and *IFN-α inducible protein 6* [*IFI6*]), which were attenuated by nirsevimab or palivizumab treatment (Fig 3D), likely because they weakened the innate immune response by reducing the amount of viral genome in respiratory epithelial cells. In contrast, although RSV infection induced the expression of *phosphoglycerate dehydrogenase* (*PHGDH*) and *phosphoserine aminotransferase 1* (*PSAT1*), which play vital roles in serine biosynthesis, nirsevimab or palivizumab treatment had no effects on their expression (Fig S4). Viral infections are known to increase *PHGDH* and *PSAT1* expression (Devadas et al, 2016), but how they impact pulmonary functions remains unclear. Although nirsevimab or palivizumab treatment can efficiently eliminate RSV, further investigations are needed to determine the consequences of altered host gene expression, including *PHGDH* and *PSAT1*.

Most of the RSV entering the respiratory tract first encounter alveolar macrophages, known to play a crucial role in the early response to RSV infection (Pribul et al, 2008; Reed et al, 2008; Eichinger et al, 2015). Therefore, we specifically examined alveolar macrophages in respiratory organoids. The expression of genes related to "innate immune response" and "immune response" was increased in the alveolar macrophage of respiratory organoids (Fig S5), and similar results could be observed in the alveolar macrophages in bronchoalveolar lavage fluid collected from RSV-infected patients. Therefore, respiratory organoids can be used to evaluate the function of alveolar macrophages during the early response to RSV infections.

# Discussion

In this study, we demonstrated that RSV replicates efficiently in respiratory organoids and can be used to evaluate disruptions in the respiratory epithelial cell layer, collagen accumulation, and innate immune, and inflammatory responses. In addition, using our respiratory organoids, although we confirmed the antiviral effects of antibodies used to prevent RSV infections, the antiviral effects of ribavirin, used as an RSV therapeutic drug, were not observed. These results suggest that our respiratory organoids are a valuable resource for elucidating the pathophysiology of RSV and advancing the development of therapeutic and preventive drugs.

RSV vaccines, such as Arexvy and Abrysvo, use recombinant RSV prefusion (preF) protein as antigens. Antibodies specific to the preF protein, formed during the fusion process of RSV and host cell membrane, are known to have high RSV-neutralizing activity (Ngwuta et al, 2015, Simões et al, 2022). By conducting neutralization tests on respiratory organoids using patient serum after

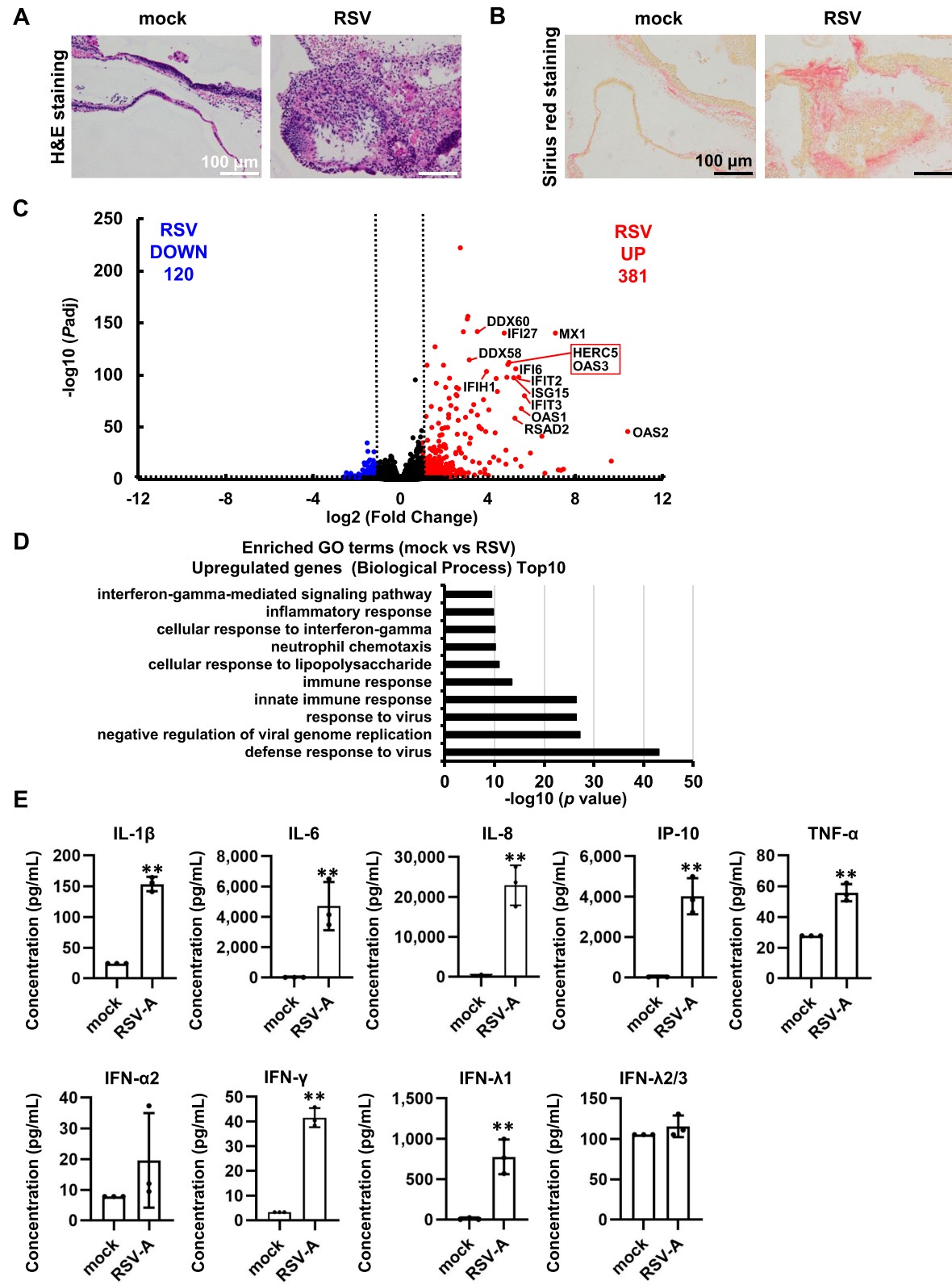

**Figure 2. Analysis of host cell response in RSV-infected respiratory organoids.**

Human iPS cell–derived respiratory organoids were infected with 0.1 TCID/cell RSV (8 × 10⁴ TCID/well) and cultured for 4 d. **(A)** H&E staining of respiratory organoids. **(B)** Sirius red staining of respiratory organoids. **(C)** A volcano plot of differentially expressed genes in uninfected and infected respiratory organoids (Log₂ fold-change > 1, adjusted *P*-value [*P*adj] < 0.01). Red and blue dots represent up-regulated and down-regulated genes, respectively. **(D)** DAVID-based GO analysis of RSV-infected

vaccination, it may be possible to evaluate vaccine efficacy and persistence more accurately. Given the importance of mucus in influencing antibody behavior within the respiratory tract (Cone, 2009), future research should also focus on developing human iPS cell–derived respiratory organoids that incorporate a functional mucus layer.

Inflammatory cytokines are deeply involved in pathogenesis caused by RSV infections (Graham et al, 2000; Zeng et al, 2011). To effectively treat RSV, it will be necessary to develop not only antiviral drugs but also develop anti-inflammatory drugs. The respiratory organoids used in this study contain macrophages, allowing us to assess accurately the release of inflammatory cytokines from macrophages. However, because immune cells other than macrophages, such as T cells and neutrophils, are also involved in severe inflammation (Geerdink et al, 2015; Karki & Kanneganti, 2021), it will be essential to construct a model incorporating those cells to more faithfully reproduce RSV pathophysiology. Furthermore, such a model would also be advantageous for developing drugs to treat respiratory dysfunction caused by RSV.

RSV is known to be more severe in infants and the elderly (Nowalk et al, 2022; Mazur et al, 2024). Most human iPS cell–derived cells are known to display fetal-type characteristics (Chen et al, 2017; Porotto et al, 2019), which is also true for the human iPS cell–derived respiratory organoids used here. Therefore, it is difficult to accurately evaluate the impact of RSV infection in the elderly using the current iteration of our human iPS cell–derived respiratory organoids. It is known that adult-type airway and lung organoids can also be generated using human somatic stem cells (Sachs et al, 2019), such as basal stem cells and AT2 cells (Katsura et al, 2020), instead of using human pluripotent stem cells such as human iPS cells. In addition, adult-type airway tissue models can be generated through air-liquid interface culture of human adult airway epithelial cells (Zhang et al, 2002; Rajan et al, 2021). These adult-type airway and lung tissue models have also proven effective in RSV research (Zhang et al, 2002; Sachs et al, 2019; Rajan et al, 2021). However, obtaining a large number of human airways and alveolar epithelial cells from diverse donors is challenging. Therefore, in addition to adult-type airway and lung tissue models, our human iPS cell–derived respiratory organoids would be valuable for efficiently conducting RSV research.

# Materials and Methods

### Human iPS cells

The human iPS cell line, 1383D6 (provided by Dr. Masato Nakagawa, Kyoto University), was maintained on 0.5 $\mu$g/cm$^2$ recombinant human laminin 511 × 10$^8$ fragments (iMatrix-511, Cat# 892 012; Nippi) with StemFit AK02N medium (Cat# RCAK02N; Ajinomoto Healthy Supply). Cell passage was performed every 6 d. For cell passaging, iPS cell colonies were treated with TrypLE Select Enzyme (Cat#

12563029; Thermo Fisher Scientific) for 10 min at 37°C and seeded with StemFit AK02N medium containing 5 $\mu$M Y-27632 (Cat# 034-24024; FUJIFILM Wako Pure Chemical).

### Respiratory organoids

To start the differentiation, human iPS cell colonies were treated with TrypLE Select Enzyme (Cat# 12563029; Thermo Fisher Scientific) for 10 min at 37°C. After centrifugation, cells were seeded onto Matrigel Growth Factor Reduced Basement Membrane (Cat# 354230; Corning)-coated cell culture plates (2.0 × 10$^5$ cells/4 cm$^2$) and cultured for 2 d. The differentiation of the respiratory organoids was performed in serum-free differentiation (SFD) medium, composed of DMEM/F12 (3:1) (Cat# 044-29765; FUJIFILM Wako Pure Chemical and Cat# 11320033; Thermo Fisher Scientific) supplemented with N2 (Cat# 141-08941; FUJIFILM Wako Pure Chemical), B-27 supplement minus vitamin A (Cat# 12587001; Thermo Fisher Scientific), ascorbic acid (50 $\mu$g/ml, Cat# ST-72132; STEMCELL Technologies), 1× GlutaMAX (Cat# 35050-061; Thermo Fisher Scientific), 1% monothioglycerol (Cat# 195-15791; FUJIFILM Wako Pure Chemical), 0.05% BSA (Cat# 820024; Sigma-Aldrich), and 1× penicillin/streptomycin). During days 0–1 of differentiation, cells were cultured with SFD medium supplemented with 10 $\mu$M Y-27632 (Cat# 034-24024; FUJIFILM Wako Pure Chemical) and 100 ng/ml recombinant activin A (Cat# 338-AC-01M; R&D Systems). During days 1–3 of differentiation, cells were cultured with SFD medium supplemented with 10 $\mu$M Y-27632 (Cat# 034-24024; FUJIFILM Wako Pure Chemical), 100 ng/ml recombinant Activin A (Cat# 338-AC-01M; R&D Systems) and 1% FBS. Between days 3–5 of differentiation, cells were cultured in SFD medium supplemented with 1.5 $\mu$M dorsomorphin dihydrochloride (Cat# 047-33763; FUJIFILM Wako Pure Chemical) and 10 $\mu$M SB431542 (Cat# 037-24293; FUJIFILM Wako Pure Chemical) for 24 h, and then SFD medium supplemented with 10 $\mu$M SB431542 and 1 $\mu$M IWP2 (Cat# 04-0034; Stemolecule) for another 24 h. During days 5–12 of differentiation, cells were cultured with SFD medium supplemented with 3 $\mu$M CHIR99021 (Cat# 034-23103; FUJIFILM Wako Pure Chemical), 10 ng/ml human FGF10 (Cat# AF-100-26; PeproTech), 10 ng/ml human FGF7 (Cat# AF-100-19; PeproTech), 10 ng/ml human BMP4 (Cat# 120-05ET; PeproTech), 20 ng/ml human EGF (Cat# AF-100-15; PeproTech), and all-trans retinoic acid (Cat# R2625 ATRA; Sigma-Aldrich). On day 12 of differentiation, cells were dissociated and embedded in the Matrigel Growth Factor–Reduced Basement Membrane to generate organoids. During days 12–20 of the differentiation, organoids were cultured in SFD medium containing 3 $\mu$M CHIR99021, 10 ng/ml human FGF10, 10 ng/ml human FGF7, 10 ng/ml human BMP4, and 50 nM ATRA. On day 20 of differentiation, organoids were recovered from the Matrigel, and the resulting suspension of organoids (small free-floating clumps) was seeded onto Matrigel-coated cell culture plates. During days 20–30 of differentiation, organoids were cultured in SFD medium containing 50 nM dexamethasone (Cat# S1322; Selleck Chemicals), 0.1 mM 8-bromo-cAMP (Cat# 1140/50; Tocris

respiratory organoids was performed. The top 10 significantly enriched GO terms of up-regulated genes (Biological Process) in RSV-infected samples compared with uninfected samples are shown. **(E)** The concentration (pg/ml) of 9 cytokines in the cell culture supernatant of the RSV-infected respiratory organoids, as measured by bead-based immunoassays. Data information: data are shown as means ± SD ($n$ = 3). Two-tailed $t$ test (**$P$ < 0.01).

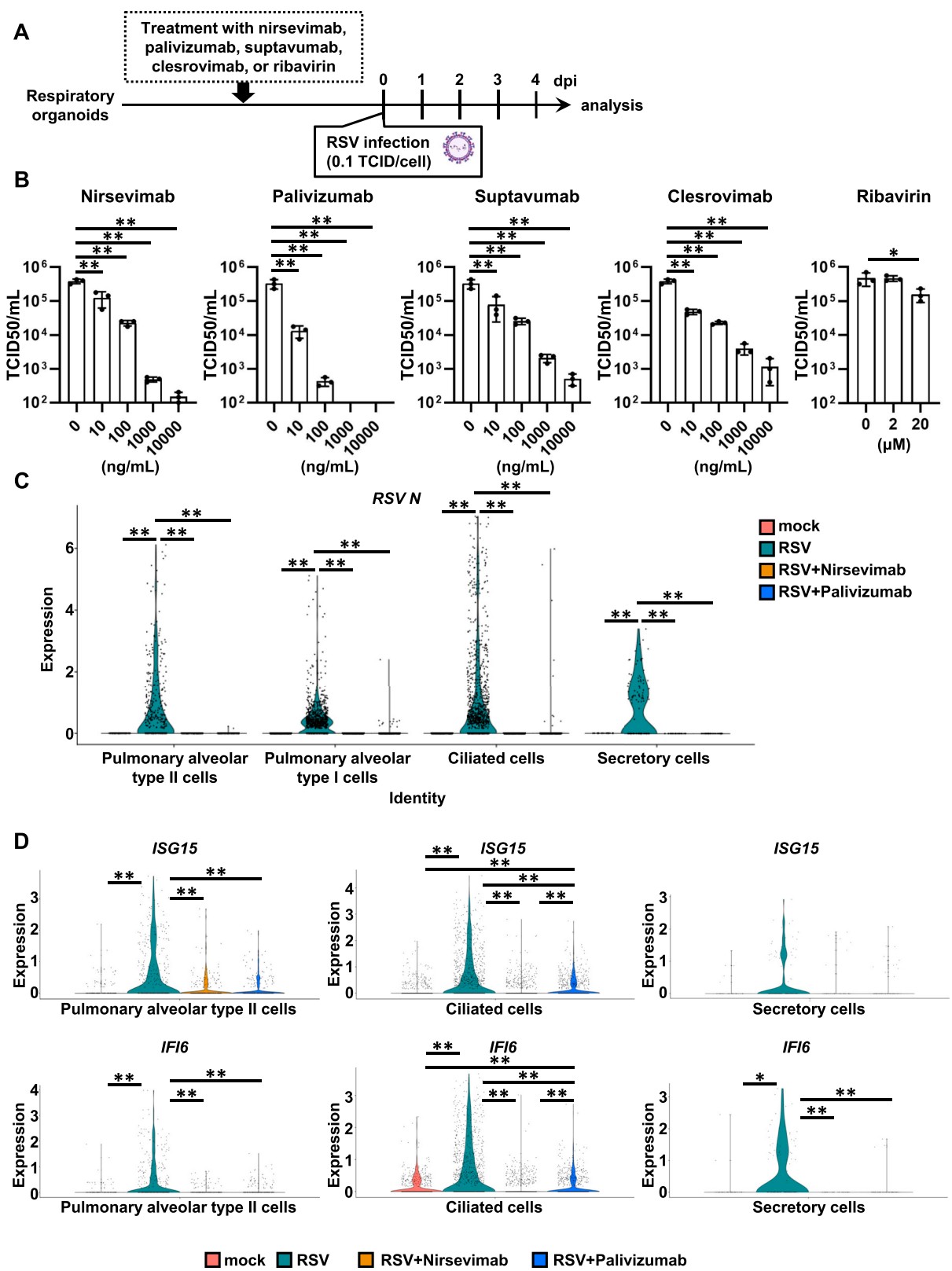

**Figure 3. Evaluation of anti-RSV F antibodies and ribavirin using human iPS cell–derived respiratory organoids.**

**(A)** Human iPS cell–derived respiratory organoids were infected with 0.1 TCID/cell (8 × 10⁴ TCID/well) RSV-A and cultured with medium containing nirsevimab, suptavumab, clesrovimab, palivizumab, or ribavirin for 4 d. **(B)** At 4 dpi, the cell culture supernatant of infected respiratory organoids was collected and TCID50 assay was performed. One-way ANOVA followed by the Dunnet post hoc test (*$P < 0.05$, **$P < 0.01$, compared with 0 ng/ml or 0 μM). Data are shown as mean ± SD ($n = 3$). **(C)** Violin plot

Bioscience), and 0.1 mM IBMX (3-isobutyl-1-methylxanthine) (Cat# 095-03413;FUJIFILM Wako Pure Chemical).

## RSV

Isolated RSV (RSV/Sendai/28-30 [subgroup A]) was used in this study (GISAID accession number: EPI_ISL_19696283). RSV was replicated in HEp-2 cells (CCL-23; ATCC) and stored at −80°C. HEp-2 cells were cultured with EMEM (Cat# 051-07615; FUJIFILM Wako Pure Chemical) supplemented with 10% FBS and 1% penicillin/streptomycin.

## Viral titration

Viral titers were measured by a median tissue culture infectious dose (TCID50) assay. Cells were seeded into 96-well cell culture plates (Cat# 167008; Thermo Fisher Scientific). Samples were serially diluted 10-fold from $10^{-1}$ to $10^{-8}$ in cell culture medium, transferred onto the cells, and incubated at 37°C for 96 h. Cytopathic effects were evaluated under a microscope. TCID50/mL was calculated using the Reed-Muench method.

## Antiviral drug and antibody assay using respiratory organoids

The human iPS cell–derived respiratory organoids were infected with 0.1 TCID50/cell ($8 \times 10^4$ TCID/well) RSV. Infected respiratory organoids were cultured with the medium containing serially diluted nirsevimab (Cat# MA5-42214; Thermo Fisher Scientific), suptavumab (Cat# HY-P99586; MedChemExpress), clesrovimab (Cat# HY-P99804; MedChemExpress), palivizumab (Cat# HY-P9944; MedChemExpress), or ribavirin (Cat# 182-02331; FUJIFILM Wako Pure Chemical). At 96 h after the infection, TCID50 assay was performed.

## Immunofluorescence staining

For immunofluorescence staining of respiratory organoids, cells were fixed with 4% PFA in PBS at 4°C. Respiratory organoids were harvested and prepared as paraffin sections (~15 $\mu$m). Paraffin was removed using xylene and afterward rehydrated with different percentages of ethanol. Antigen retrieval was performed with 0.1%-tTBS (10×) (pH 7.4) (Cat# 12750-81; Nacalai Tesque). For blocking non-specific staining, the slides were incubated in Blocking One (Cat# 03953-066; Nacalai Tesque) for 10 min at room temperature. Primary antibody incubation was performed overnight at 4°C. Stained slides were washed thrice with 1× PBS (Cat# 14249-24; Nacalai Tesque) the next day. Alexa Fluor 488- or 594-conjugated secondary antibody incubation was performed at room temperature for 45 minutes, and slides were washed afterward three times for 5 min. Sections were washed, mounted with ProLong Glass Antifade Mountant with NucBlue Stain (Cat# P36985; Thermo Fisher Scientific) and DAPI (Cat# 12593-64; Nacalai Tesque), and analyzed using an inverted laser scanning confocal microscopy system (FV3000; Evident). Antibodies used are summarized in Table S3.

## Measurement of cytokines in cell culture supernatants

For measuring cytokine concentrations, the cell culture supernatants of respiratory organoids were analyzed using the Human Anti-Virus Response Panel (Cat# 740349; BioLegend). Flow cytometry was performed according to the manufacturer's instructions using MACSQuant Analyzer 10 Flow Cytometer (Miltenyi Biotec).

## Hematoxylin and eosin and sirius red staining

Respiratory organoids were fixed with 4% PFA (Cat# 163-20145; FUJIFILM Wako Pure Chemical) for 15 min, harvested, and used to prepare paraffin sections. Paraffin-embedded tissue sectioning and histological staining were performed by the Applied Medical Research Laboratory.

## Capillary-based immunoassay

Respiratory organoids were lysed in RIPA buffer (Cat# 89900; Thermo Fisher Scientific) containing a protease inhibitor mixture (Cat# P8340; Sigma-Aldrich). After the centrifugation, the supernatants were collected. Antibody-based protein quantification was performed using a Jess system (ProteinSimple) with the 12–230 kD Separation Module (Cat#SM-W001; ProteinSimple) as instructed. Antibodies used for this analysis are summarized in Table S3. Data were analyzed and visualized using Compass for Simple Western software (ProteinSimple). Jess protein images are shown throughout the manuscript.

## Ultrathin section transmission electron microscopy

Respiratory organoids were fixed with 2.5% glutaraldehyde in 0.1M cacodylate buffer and post-fixed with 1% osmium tetroxide in the same buffer at 4°C. Then, they were dehydrated in a series of ethanol gradients and embedded in epoxy resin. Ultrathin sections were cut, stained with uranyl acetate and lead citrate, and examined using an electron microscope (HT-7700; Hitachi High-Tech Corporation) at 80 kV.

## Bulk RNA sequencing

Total RNA was isolated from RSV-infected respiratory organoids. RNA integrity was assessed with a 2100 Bioanalyzer (Agilent Technologies). RNA-seq libraries were constructed using a TruSeq Stranded mRNA Sample Prep Kit (Illumina) according to the manufacturer's instructions. Sequencing was performed on an Illumina NextSeq 2000 with the paired-end mode. FASTQ files were generated using bcl2fastq-2.20. Adapter sequences and low-quality

displaying the gene expression of *RSV-A N* in each cluster of epithelial cells (alveoloar type II cells, alveolar type I cells, ciliated cells, and secretory cells). **(D)** Violin plot displaying the gene expression of *interferon stimulated gene* (*ISG*) *15* and *interferon alpha inducible protein* (*IFI*) *6* in alveolar type II cells, ciliated cells, and secretory cells of uninfected and RSV-infected respiratory organoids treated with or without antibodies (nirsevimab or palivizumab). Data information: Wilcoxon rank-sum test with the Bonferroni correction. (**P < 0.01).

bases were trimmed from the raw reads using cutadapt ver v4.1 (Kechin et al, 2017). Trimmed reads were mapped to human reference genome sequences (hg38) using STAR ver 2.7.10a (Dobin et al, 2013) with GENCODE (release 32, GRCh38.p13) (Frankish et al, 2019) GTF file. The uniquely and properly mapped reads were used for further analysis. Raw read counts were calculated using htseq-count ver. 2.0.2 (Anders et al, 2015) with the GENCODE GTF file. Differential expression analysis was performed by DESeq2 v1.34.0 (Love et al, 2014) using the Wald test. The $P$-values were corrected for multiple testing using the Benjamini and Hochberg method and represented as the false discovery rate. Raw data from this study were submitted to the Gene Expression Omnibus (GEO) under accession number GSE263272.

### scRNA-seq

Respiratory organoids were dissociated, and the cell concentration was adjusted to 1,000 cells/$\mu$l. The single-cell suspension, with a concentration of 1,200 cells/$\mu$l, was loaded onto a 10x Genomics Chromium Next GEM Single Cell G chip (10x Genomics) with enzyme mix, gel beads, and oils to generate gel beads-in-emulsion (Chromium Next GEM single cell 3' kit v3.1) (10x Genomics). Subsequently, reverse transcription and cDNA amplification were conducted according to the manufacturer's instructions using Verity thermal cycler (Thermo Fisher Scientific) for the Chromium Next GEM Single Cell kit. After adding index sequences to the library pool, the cDNA library was sequenced on the Nova-seq X platform (Illumina).

The 10x Genomics Cell Ranger pipeline (version 7.1.0) was used to perform sample demultiplexing, alignment to the GRCh38-2020-A human reference genome, barcode/unique molecular identifiers (UMI) processing, and gene counting for each cell. Gene count matrix data were processed as follows. Briefly, the low-quality cells were excluded based on the following criteria: (I) number of detected genes ≤ 200, (II) number of UMI molecules detected within a cell ≥ 100,000, and (III) percentage of mitochondrial genes expressed ≤ 10%. Filtered UMI feature-barcode matrices were processed with ICARUS v.2.0 (Jiang et al, 2023). Gene count matrices were normalized and scaled by 10,000. Dimensionality reduction with principal component analysis was performed using a set of 2,000 top variable genes (Seurat::FindVariableFeatures) (Hao et al, 2024). Clustering was performed with 50 principal components, a k-nearest neighbor value of 50, and the Louvain clustering algorithm. Cell type annotation was performed by scType (Ianevski et al, 2022). scRNA-seq expression datasets of bronchoalveolar lavage fluid of RSV-infected patient (Travaglini et al, 2020) were obtained from a public database. The accession numbers for scRNA-seq data obtained in this study (RSV+nirsevimab and RSV+palivizumab) are GSE263271. The accession numbers for scRNA-seq data (mock and RSV) are GSE260752.

### Statistical analyses

Statistical significance was evaluated using an unpaired two-tailed $t$ test, one-way analysis of variance (ANOVA) followed by Dunnet post hoc tests, or Wilcoxon rank-sum test with the Bonferroni correction. Statistical analyses were performed using GraphPad Prism 9. Data are representative of three independent experiments. Details are described in the figure legends.

## Data availability

Bulk RNA-seq data from this study were submitted to the Gene Expression Omnibus (GEO) under accession number GSE263272 (reviewer token: efolkkyanvobjst). The accession numbers for scRNA-seq data (RSV+nirsevimab, and RSV+palivizumab) obtained in this study are GSE263271 (reviewer token: cvabgwqytlkbvst). Detailed information including genomic sequence of isolated RSV (RSV/Sendai/28-30 [subgroup A]) was submitted to GISAID under accession number EPI_ISL_19696283.

## Supplementary Information

## Acknowledgements

We thank Dr. Kelvin Hui (Kyoto University) for critical reading of the manuscript, Ms. Kazumi Deguchi and Ms. Satoko Sakurai (Kyoto University) for technical assistance with the RNA-seq experiments, and Ms. Natsumi Mimura (Kyoto University) for technical assistance. This research was supported by the iPS Cell Research Fund, the Japan Agency for Medical Research and Development (AMED) (JP21gm1610005, JP23fk0210135, JP23bm1323001, JP23jf0126002, JP243fa627001), JSPS Core-to-Core Program (A. Advanced Research Networks, JPJSCCA20240006), and the Joint Usage/Research Center Program of the Institute for Life and Medical Sciences at Kyoto University.

### Author Contributions

R Hashimoto: resources, data curation, formal analysis, investigation, methodology, and writing—original draft.
Y Watanabe: formal analysis, visualization, and writing—original draft.
A Keshta: data curation and investigation.
M Sugiyama: data curation and investigation.
Y Kitai: resources and investigation.
A Hirabayashi: data curation and investigation.
N Yasuhara: data curation and investigation.
S Morimoto: data curation and investigation.
A Sakamoto: formal analysis and investigation.
Y Matsumura: resources and investigation.
H Nishimura: resources.
T Noda: data curation, formal analysis, and investigation.
T Yamamoto: data curation and investigation.
M Nagao: resources and investigation.
M Takeda: resources and investigation.
K Takayama: conceptualization, supervision, funding acquisition, project administration, and writing—review and editing.

## Conflict of Interest Statement

The authors declare that they have no conflict of interest.

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
