## [Reviewer comments · Life Science Alliance]

Life Science Alliance

Human iPS cell-derived respiratory organoids as a model for respiratory syncytial virus infection

Rina Hashimoto¹, Yukio Watanabe, Abeer Keshta, Masaya Sugiyama, Yuki Kitai, Ai Hirabayashi, Naoko Yasuhara, Shiho Morimoto, Ayaka Sakamoto, Yasufumi Matsumura, Hidekazu Nishimura, Takeshi Noda, Takuya Yamamoto, Miki Nagao, Makoto Takeda and Kazuo Takayama

DOI: <https://doi.org/10.26508/lsa.202402837>

Corresponding author(s): Dr. Kazuo Takayama (Kyoto University; Institute of Science Tokyo)

Review Timeline:

Submission Date:	2024-05-23
Editorial Decision:	2024-08-02
Revision Received:	2025-02-07
Editorial Decision:	2025-03-05
Revision Received:	2025-03-29
Editorial Decision:	2025-03-31
Revision Received:	2025-04-02
Accepted:	2025-04-10

Scientific Editor: Tim Fessenden

Transaction Report:

August 2, 2024

Re: Life Science Alliance manuscript #LSA-2024-02837-T

Dr. Kazuo Takayama
Kyoto University
Shogoin Kawaharacho 53, Sakyo-ku
Kyoto 6068507
Japan

Dear Dr. Takayama,

Thank you for submitting your manuscript entitled "Evaluation of therapeutic agents for respiratory syncytial virus using human respiratory organoids" to Life Science Alliance. The manuscript was assessed by expert reviewers, whose comments are appended to this letter. We invite you to submit a revised manuscript addressing the Reviewer comments.

Thank you for this interesting contribution to Life Science Alliance. We are looking forward to receiving your revised manuscript.

Sincerely,

B. MANUSCRIPT ORGANIZATION AND FORMATTING:

Reviewer #1 (Comments to the Authors (Required)):

Respiratory organoids including iPS derived respiratory organoids are a very powerful platform to study respiratory virus-host interactions at the individual level. One organoid line represents the unique genetic background of the individual from which the organoid was generated from. This manuscript reports on the effect of respiratory syncytial virus (RSV) on a single line of iPS derived respiratory organoid and its potential for evaluating therapeutics in particular monoclonal antibodies. It was clearly demonstrated that RSV infected iPS respiratory organoid, the cell populations infected with RSV by scRNA seq, the host response by bulk RNA seq and targeted mRNA and cytokines/chemokine protein concentrations, and the EC50 and EC90 based on virus gene copy numbers. However, there were a number of significant limitations that need to be addressed to strengthen the excellent data presented. In the field of RSV, it is not enough to understand viral replication and host response in terms of virus copy numbers but it is crucial to have information on the infectious virus load that is being used for the MOI, that is being generated by the infected iPS derived respiratory organoids, and to determine the EC50 and EC90 of therapeutic drugs. In addition, when comparing the biology of RSV/A and RSV/B the infectious virus load is what is used to determine the MOI and not the gene copy number, in particular, when different RT-PCR assays are used to generate the virus gene copy number. It is like comparing oranges to apples.

Other Comments.

Abstract.

Suggest revising the first two sentences because development of monoclonal antibodies and antiviral drugs have been ongoing for many years. In 2023 nirsevimab was the only mAb that was approved, what other antibody was developed? Please note that all recent RSV/B isolates are resistant to suptavumab, which is why suptavumab failed to meet its primary endpoint in its phase III clinical trial. Suptavumab was not licensed and not further developed for immunoprophylaxis. Likewise, aerosol ribavirin, although approved for use, is rarely used except by some institution in the immunocompromised setting.

Introduction.

Lines 68-70. Please revise this sentence. The MMWR showed a higher disease severity but substantially lower hospitalization numbers in older adults compared to SARS-CoV-2 and flu during the study period.

Lines 73-77. Suggest revising. Aerosol ribavirin is rarely used to treat RSV infection in the U.S. and Europe. Currently, there is no approved antiviral drug that is safe and effective. Agree there is a significant need for developing safe and effective antiviral drugs for the treatment of RSV infection in both the adult and pediatric populations.

The introduction can get confusing when the first paragraph talks about older adults for which vaccines are now available and this was not mentioned.

The second paragraph focuses on monoclonal antibodies, which are approved only for a subset of preterm or high-risk infants (palivizumab) or all infants under 1 year of age (nirsevimab), and a subset of high-risk infants in their second RSV season. As currently written, it suggest mAbs are also for older adults. Please revise the paragraph so that it is clear that mAbs are approved for the prevention of severe RSV infection in infants and not older adults.

Lines 89-91 statement needs a reference.

Results. It is clear that RSV/A infected iPS cell-derived respiratory organoids, however, information on the production of infectious viruses was not demonstrated. For truly studying antiviral agents and mAbs and determining EC90s, the primary measure needs to be infectious virus. A quantitative plaque assay is usually done to quantitate the amount of infectious viruses. Although gene copy numbers is an alternate method, it does not reflect well the production of infectious viruses, especially when evaluating antiviral agents. Please include data on the amount of infectious viruses being produced by the respiratory organoids. Also, more detail information on the RSV strains used are needed. When were the viruses isolated, to what genotype do they belong, what is their passage history (lab strain or clinical strain). Have the sequences of these RSV strains been deposited in GenBank? They should be. Please provide the accession numbers.

Figure 1g. Recommend listing some of the top regulated and down regulated genes in the volcano plot.

Also, what is not cleared and should be provided in the method section is the method used to infect the respiratory organoids. Was the virus inoculum simply placed in the airway organoid media and infected via the basal cells or injected into the apical lumen to infect the ciliated cells. Also, how was the MOI determined and what was used to define the viral inoculum; viral gene copy number or an infectious unit like plaque forming unit. This will become highly relevant when RSV/A and RSV/B are being compared. In each experiment, how many respiratory organoids are used to generate the bulk transcriptomic data, virus gene copy number data and cytokine concentration?

Line 134. Suggest revising the word fibrosis to increase collagen deposition. Fibrosis of the organoid was not demonstrated.
Line 137. It is very difficult to distinguish the RSV infecting subtype (A vs. B) based on the severity of the infection. There are studies that show no difference in disease severity between the RSV subtypes or increase severity with RSV/B infection. Agree that organoids might be able to demonstrate differences in pathogenicity at the organoid level. A significant flaw in the comparison between RSV/A and RSV/B strains is how the MOI was determined. From the method section, it suggests that it was based on gene copy number and the gene used for detection was different with the N and P genes used for the detection of RSV/A and RSV/B respectively. Infectious virus concentration and not gene copy number is needed to ensure the MOI (infectious virus) used to infect the organoids for both RSV strains are comparable. The lower levels of P gene based on the CT difference to GAPDH suggest a lower level of replication that could be due to either a difference in virus inoculum or greater resistance by the respiratory organoid to RSV/B compared to RSV/A. Based on the gene expression of IFN and interferon inducible genes the later is unlikely. Also the PCA plot, in particular PC1 accounted for low variance (20.18%) between the three groups suggesting the variance between the three groups (including mock) are low and not well defined at day 4 post inoculation.

The IFIH1 knockout human iPS experiments clearly demonstrated the impact of MDA5 in the innate immune response and in reducing viral replication. This experiment would be strengthened with infectious virus data. What does doubling the gene copy number or increasing the relative protein F level by 3 folds mean in terms of infectious titer?

Evaluation of antibodies and antiviral drugs against RSV section. It would be very helpful if graphs were generated to illustrate the EC90 for the different mAbs studied. Although the EC50 and EC90 are shown in a Table (Figure 3a), a plot for each mAb at the various concentration should also be provided. As stated previously, EC50 and EC90 should also be provided for the infectious virus concentration and not just the gene copy number.

Figure 4. the dominant genes (for example top 4 unique genes) used to define the 11 cell clusters identified in the respiratory organoids by scRNA seq should be provided in the supplement section. It is interesting that AT2 cells expanded during RSV infection instead of being destroyed and that the fibroblast cell population expanded with nirsevimab and palivizumab immunoprophylaxis instead of decrease. Was there evidence of increase collagen deposition in the nirsevimab and palivizumab treated organoids? Please address.

Some of the major cell populations such as AT1 & AT2 cells, macrophages, basal cells, secretory cells should be confirmed by IF or flow data. The ciliated cell type was demonstrated but this reviewer was unable to see the cilia in the IF images stained with acetylated α -tubulin (Fig. 1c & Fig S3c).

Discussion. Line 245. To date mAbs have not worked as a therapeutic drug for RSV bronchiolitis in infants. Suggest rephrasing or include the human data that demonstrate lack of efficacy.

The authors should consider mentioning other airway organoid models used to study RSV and other respiratory viruses that are not based on iPS; perhaps a few sentences comparing and contrasting.

Reviewer #2 (Comments to the Authors (Required)):

The authors describe the use of air liquid interface organoids to describe the efficacy of prophylactic antibodies against RSV in the model. The work is interesting as I don't recall anyone else testing RSV biologics in this system. The data look good and are interesting, but the manuscript reads like a first and unrefined draft. It needs a lot of work to bring the writing up to a proper scientific standard. I outline what needs to be done but this is by and far not a complete list. I suggest the authors seek editorial support from a colleague to help write the paper or a scientific editorial service.

1. In the results sections the authors need to provide a lead-up which is justification and the background prior to outlining the results. This gives the work context. For example the CRISPR KO of MDA5 is interesting work but they provide no justification for why they do the experiments. They claim that another study, briefly noted at the end of the section, found that RIG-I was ore antiviral than MDA5 but they do no experiments examining RIG-I so there is no comparator.

2. In line 187 they state that the RSV Mabs are therapeutic. They are not, they are only licensed for prevention of RSV, they are prophylaxis. This fact needs to be reflected in the introduction as well where they introduce remdesivir. They don't need to introduce remdesivir because it is a completely different type of drug and doesn't work against RSV.

3. The abstract needs significant work. For example the first sentence is too awkward and needs to be rewritten. The results must be referred to in the past tense, they are currently referred to in the present tense. They do not encompass all of the results shown in the paper in the abstract the way it is written. The abstract makes the paper seem as though it is a much smaller study than it is. They say that they did ScSEQ to find the infected cells but this method also was used to characterize the model system. In summary, The abstract is difficult to understand, it doesn't adequately cover the contents of the paper, and needs to be rewritten.

4. In the figures, what do the authors mean by RSV-FP? Do they mean this is RSV-F and RSV-P or just RSV-F as is the norm for this protein. If it is just RSV-F then merely write 'RSV-F.' If it is P protein as well include a comma or such to distinguish the two proteins.

5. JESS analysis in the materials and methods section needs to have a new subtitle. It should be called Western blot or Capillary

based protein analysis. Something in this regard. JESS is the name of the system and not the method. The materials and methods needs more references as does the entire paper. They include catalogue numbers which is good but they don't cite the literature that first describes the method.

6. In the discussion or results the authors should compare the EC50s of the Mabs in their system compared to other systems. Are they the same, similar, more sensitive?

7. In figure 1G what are some of the genes represented by the dots? Why not outline the highest and lowest expressed genes? Otherwise it is entirely moot as to why one would do the single cell SEQ in the first place.

In summary, this paper needs a very significant amount of work. All section needs to be rewritten, the data need be highlighted more and analysed more in the results and discussed with reference to more studies in the discussion. This is not an exhaustive list.

Reviewer #3 (Comments to the Authors (Required)):

Hashimoto et al. described the usage of airway organoids to evaluate the efficacy of therapeutic agents against RSV. Airway organoids were generated from iPSC cells and differentiated to cover all cell types present in the respiratory tract, from ciliated epithelial cells to alveolar cells. The methods used to analyze RSV infection in organoids are determination of viral copies, western blot and RNA sequencing. The authors describe the efficacy of several different therapeutic agents and conclude that more therapeutic agents should be discovered. I have several issues with this manuscript.

Major comments

1. The title and abstract poorly reflect the content of the manuscript.
2. The authors discuss RSV-specific monoclonal antibodies as therapeutic agents, whereas these are licensed (and clinically effective) for prophylactic use (see manuscript line 77/82).
3. In many experiments the experimental design is unclear, the manuscript contains insufficient information to reproduce the experiments. The authors describe the differentiation of airway organoids from iPSCs using a matrigel-based method. It is not clear if the differentiation is performed in 2D or 3D. In addition, it is not clear which part of the respiratory tract these organoids should reflect, nasal, tracheal, bronchial or alveoli.
4. If the major aim was to evaluate antiviral agents for RSV in respiratory organoids, it is unclear why the authors chose RNAseq as primary read-out. It would have been much more cost-effective to use immunofluorescence staining of infected cells.
5. The introduction contains insufficient background information. Information on the background of RSV is completely lacking. The text focuses on antivirals and preventive monoclonal antibodies, until (in line 103) the authors mention for the first time that they analyzed the host response to RSV infection using organoids. The purpose of this analysis is not connected to the aim of the study. It is not clear to me (and not explained by the authors) why iPSC-derived respiratory organoids contain macrophages.
6. Results: line 109 is the first place where iPSCs are mentioned, and the first three results paragraphs focus completely on analysis of host responses. RSV subgroups are suddenly introduced in line 114, but were not mentioned in the introduction.
7. Results: the legend to figure 1 mentions that samples for RNAseq were collected four days after RSV infection. The authors do not explain how this time point was chosen, and do not include a growth curve of RSV in their iPSC-derived respiratory organoids.
8. In line 137 -157 the authors directly compare host responses to RSV A and B strains. Again, it is unclear how this relates to the objectives of the manuscript. Also, it is unclear if differences between two strains can be translated into differences between subgroups.
9. In lines 161-178 the authors try to elucidate the mechanism of RSV-specific innate response. Again, it is unclear how this relates to the objectives of the manuscript.
10. Results: in lines 196-237 the authors describe they performed RNAseq analysis of RSV infected organoids treated with nirsevimab or palivizumab. The aim of this experiment is completely unclear to me. As described by the authors (cited in my major comment 2), these monoclonal antibodies were developed for prophylactic use, not for therapeutic use.
11. The discussion completely lacks comparison with the international literature, either in comparison with other organoid models for RSV, other RNAseq studies for RSV or other model systems used for evaluation of antivirals or antibodies to RSV.

Instead, the authors have included references of the literature in their results section. I would strongly recommend to keep results and discussion separate. The authors should discuss their model systems with models for culture of differentiated epithelial cells at air-liquid interface.

Specific comments

12. Title: remove 'therapeutic' and add 'iPSC-derived'

13. Running title: remove 'lung' (no lung tissue was used)

14. Abstract: remove EC90 data.

15. In line 53 instead of 'them' rather write RSV

16. Type in line 80, change nirsavimab to nirsevimab

17. Figure 1C: it is unclear to me what is stained by the antibody to acetylated-alpha-tubulin, but it does not look like cilia. It would be useful to match the RNAseq data with convincing immunofluorescence staining of the different cell types in the organoid cultures.

18. Figure 1D: I do not understand the added value of the TEM data. The circular particle shape is concerning, as RSV particles are considered to be filamentous in vivo.

19. It is not specified why the N protein for RSV-A is measured, but the P protein for RSV-B for figure S3.

20. Lines 193-196 (and tables): the authors should reconsider the precision of their results. I find it unrealistic to report these with two digits behind the comma.

21. Line 369: specify passage number. Is this virus adapted to in vitro cell culture?

The authors describe there is room for improvement in the development of antiviral drugs. However, no new compounds are tested and the existing ones have already been tested on airway organoids/HAE cells.

The authors claim their model to be useful for testing therapeutic agents against RSV. However, a big downside of organoid models is that they are costly and time-consuming. The paper would be stronger if the authors describe why this model is so much better over the now often used immortalized cell lines, and how this model can be used in high-throughput to be able to test all these different therapeutic agents.

Dr. Eric Sawey
Executive Editor
Life Science Alliance

February 7, 2025

Dear Dr. Sawey,

Thank you for reviewing our manuscript, "Human iPS cell-derived respiratory organoids as a model for respiratory syncytial virus infection" (#LSA-2024-02837-T). The comments from the editor and reviewers were very thoughtful and instructive. Each comment has been addressed below in a point-wise manner. We have modified the manuscript according to all the comments and suggestions. We believe the manuscript has vastly improved and is now ready for publication. The sentences we modified in the revised manuscript are in red.

Sincerely,

Kazuo Takayama, Ph.D.
Center for iPS Cell Research and Application (CiRA), Kyoto University, Shogoin
Kawaharacho 53, Sakyo-ku, Kyoto 606-8507, Japan
Tel: +81-75-366-7362
Fax: +81-75-366-7098
E-mail: kazuo.takayama@cira.kyoto-u.ac.jp

A point-by-point response to reviewer's comments

Reviewer #1

Comment 1. Respiratory organoids including iPS derived respiratory organoids are a very powerful platform to study respiratory virus-host interactions at the individual level. One organoid line represents the unique genetic background of the individual from which the organoid was generated from. This manuscript reports on the effect of respiratory syncytial virus (RSV) on a single line of iPS derived respiratory organoid and its potential for evaluating therapeutics in particular monoclonal antibodies. It was clearly demonstrated that RSV infected iPS respiratory organoid, the cell populations infected with RSV by scRNA seq, the host response by bulk RNA seq and targeted mRNA and cytokines/chemokine protein concentrations, and the EC50 and EC90 based on virus gene copy numbers. However, there were a number of significant limitations that need to be addressed to strengthen the excellent data presented. In the field of RSV, it is not enough to understand viral replication and host response in terms of virus copy numbers but it is crucial to have information on the infectious virus load that is being used for the MOI, that is being generated by the infected iPS derived respiratory organoids, and to determine the EC50 and EC90 of therapeutic drugs. In addition, when comparing the biology of RSV/A and RSV/B the infectious virus load is what is used to determine the MOI and not the gene copy number, in particular, when different RT-PCR assays are used to generate the virus gene copy number. It is like comparing oranges to apples.

Response: We appreciate the reviewer's comments. In the revised manuscript, viral replication was evaluated using TCID50 assay instead of viral copy numbers. Figure 1B presents TCID50 values over time. Similarly, anti-RSV F antibody evaluation experiments were conducted using TCID50 assay rather than measuring viral copy numbers (Fig. 3B).

[Figures removed by editorial staff per authors' request]

Comment 2. Abstract. Suggest revising the first two sentences because development of monoclonal antibodies and antiviral drugs have been ongoing for many years. In 2023 nirsevimab was the only mAb that was approved, what other antibody was developed? Please note that all recent RSV/B isolates are resistant to suptavumab, which is why suptavumab failed to meet its primary endpoint in its phase III clinical trial. Suptavumab was not licensed and not further developed for immunoprophylaxis. Likewise, aerosol ribavirin,

although approved for use, is rarely used except by some institution in the immunocompromised setting.

Response: As noted by the reviewer, the descriptions of anti-RSV F antibodies and antiviral drugs were inaccurate. The abstract has been modified. Please see lines 50-65.

Comment 3. Introduction. Lines 68-70. Please revise this sentence. The MMWR showed a higher disease severity but substantially lower hospitalization numbers in older adults compared to SARS-CoV-2 and flu during the study period.

Response: Thank you for your comment. We have revised the manuscript. Please see lines 67-72.

Comment 4. Lines 73-77. Suggest revising. Aerosol ribavirin is rarely used to treat RSV infection in the U.S. and Europe. Currently, there is no approved antiviral drug that is safe and effective. Agree there is a significant need for developing safe and effective antiviral drugs for the treatment of RSV infection in both the adult and pediatric populations.

Response: The description of ribavirin has been revised in the manuscript. Please see lines 88-92.

Comment 5. The introduction can get confusing when the first paragraph talks about older adults for which vaccines are now available and this was not mentioned. The second paragraph focuses on monoclonal antibodies, which are approved only for a subset of preterm or high-risk infants (palivizumab) or all infants under 1 year of age (nirsevimab), and a subset of high-risk infants in their second RSV season. As currently written, it suggest mAbs are also for older adults. Please revise the paragraph so that it is clear that mAbs are approved for the prevention of severe RSV infection in infants and not older adults.

Response: In the revised manuscript, the description of monoclonal antibodies has been corrected. Please see lines 92-98.

Comment 6. Lines 89-91 statement needs a reference.

Response: We have added the reference. Reference: Mullard A, 2023 (lines 97-98).

Comment 7. Results. It is clear that RSV/A infected iPS cell-derived respiratory organoids, however, information on the production of infectious viruses was not demonstrated. For truly studying antiviral agents and mAbs and determining EC90s, the primary measure needs to be infectious virus. A quantitative plaque assay is usually done to quantitate the amount of

infectious viruses. Although gene copy numbers is an alternate method, it does not reflect well the production of infectious viruses, especially when evaluating antiviral agents. Please include data on the amount of infectious viruses being produced by the respiratory organoids. Also, more detail information on the RSV strains used are needed. When were the viruses isolated, to what genotype do they belong, what is their passage history (lab strain or clinical strain). Have the sequences of these RSV strains been deposited in GenBank? They should be. Please provide the accession numbers.

Response: To confirm that RSV-A infected iPSC-derived respiratory organoids, we conducted TCID₅₀ assay (**Fig. 1B**). Another TCID₅₀ assay was also performed in the experiment evaluating anti-RSV F antibodies (**Fig. 3B**). Details regarding the RSV strain used in this study have been added to the Methods section. The RSV strain belongs to genotype A and was isolated from an RSV-infected patient at Sendai Medical Center. It was passaged twice in HEp-2 cells, and its sequence has been deposited in GISAID. Accession number: EPI_ISL_19696283. Please see lines 306-307.

Comment 8. Figure 1g. Recommend listing some of the top regulated and down regulated genes in the volcano plot.

Response: The genes with altered expression are highlighted in the volcano plot (**Fig. 2C**).

[Figure removed by editorial staff per authors' request]

Comment 9. Also, what is not cleared and should be provided in the method section is the method used to infect the respiratory organoids. Was the virus inoculum simply placed in the airway organoid media and infected via the basal cells or injected into the apical lumen to infect the ciliated cells. Also, how was the MOI determined and what was used to define the viral inoculum; viral gene copy number or an infectious unit like plaque forming unit. This will become highly relevant when RSV/A and RSV/B are being compared. In each experiment, how many respiratory organoids are used to generate the bulk transcriptomic data, virus gene copy number data and cytokine concentration?

Response: The infection experiment method is illustrated in **Figure 1A** and described in the manuscript. A separate study on the development of respiratory organoids is currently under review in another journal. Since respiratory organoids exhibit an apical-out structure, efficient infection experiments can be conducted by simply adding the virus to the culture medium. In this study, inoculations were performed at a multiplicity of 0.1 TCID₅₀/cell, with TCID values determined using HEp-2 cells. Although

approximately 8×10^5 cells can be recovered from a single well, the exact organoid count cannot be determined due to the repeated budding of organoids. For bulk RNA-seq, viral copy number analysis, and cytokine concentration measurements, organoids from one well (8×10^5 cells/well) were used.

[Figure removed by editorial staff per authors' request]

Comment 10. Line 134. Suggest revising the word fibrosis to increase collagen deposition. Fibrosis of the organoid was not demonstrated.

Response: We have revised the manuscript according to your suggestions. Please see lines 138-140.

Comment 11. Line 137. It is very difficult to distinguish the RSV infecting subtype (A vs. B) based on the severity of the infection. There are studies that show no difference in disease severity between the RSV subtypes or increase severity with RSV/B infection. Agree that organoids might be able to demonstrate differences in pathogenicity at the organoid level. A significant flaw in the comparison between RSV/A and RSV/B strains is how the MOI was determined. From the method section, it suggests that it was based on gene copy number and the gene used for detection was different with the N and P genes used for the detection of RSV/A and RSV/B respectively. Infectious virus concentration and not gene copy number is needed to ensure the MOI (infectious virus) used to infect the organoids for both RSV strains are comparable. The lower levels of P gene based on the CT difference to GAPDH suggest a lower level of replication that could be due to either a difference in virus inoculum or greater resistance by the respiratory organoid to RSV/B compared to RSV/A. Based on the gene expression of IFN and interferon inducible genes the later is unlikely. Also the PCA plot, in particular PC1 accounted for low variance (20.18%) between the three groups suggesting the variance between the three groups (including mock) are low and not well defined at day 4 post inoculation.

Response: As you suggested, the data in the previous version did not allow for an accurate comparison between RSV-A and RSV-B. Since it is not possible to directly measure the RSV titers in iPSC-derived respiratory organoids, we first determine the TCID values using HEp-2 cells and subsequently perform infection experiments in organoids at 0.1 TCID/cell. As a result, achieving the same MOI for RSV-A and RSV-B in iPSC-derived respiratory organoids is nearly impossible. Therefore, in the revised manuscript, we have decided not to compare RSV-A and RSV-B.

Comment 12. The IFIH1 knockout human iPS experiments clearly demonstrated the impact of MDA5 in the innate immune response and in reducing viral replication. This experiment would be strengthened with infectious virus data. What does doubling the gene copy number or increasing the relative protein F level by 3 folds mean in terms of infectious titer?

Response: We have decided to publish the data on MDA5 in a separate paper, as we believe this manuscript should focus on evaluating anti-RSV F antibodies using respiratory organoids.

Comment 13. Evaluation of antibodies and antiviral drugs against RSV section. It would be very helpful if graphs were generated to illustrate the EC90 for the different mAbs studied. Although the EC50 and EC90 are shown in a Table (Figure 3a), a plot for each mAb at the various concentration should also be provided. As stated previously, EC50 and EC90 should also be provided for the infectious virus concentration and not just the gene copy number.

Response: To evaluate anti-RSV F antibodies using respiratory organoids, we conducted a TCID50 assay (Fig. 3B) and measured the TCID50 values at different concentrations of anti-RSV F antibodies.

Comment 14. Figure 4. the dominant genes (for example top 4 unique genes) used to define the 11 cell clusters identified in the respiratory organoids by scRNA seq should be provided in the supplement section. It is interesting that AT2 cells expanded during RSV infection instead of being destroyed and that the fibroblast cell population expanded with nirsevimab and palivizumab immunoprophylaxis instead of decrease. Was there evidence of increase collagen deposition in the nirsevimab and palivizumab treated organoids? Please address.

Response: scType was used for cell type annotation, and the list of genes used for this purpose is summarized in Table S2. Fibroblast markers were not elevated in the respiratory organoids treated with nirsevimab and palivizumab, and collagen deposition remained unchanged (data not shown). Given the difficulty in drawing definitive conclusions regarding fibrosis from the current data, this point was omitted from the revised manuscript.

Comment 15. Some of the major cell populations such as AT1 & AT2 cells, macrophages, basal cells, secretory cells should be confirmed by IF or flow data. The ciliated cell type was demonstrated but this reviewer was unable to see the cilia in the IF images stained with acetylated α -tubulin (Fig. 1c & Fig S3c).

Response: The characteristics of respiratory organoids are analyzed in a separate study, which is currently under review in another journal. The manuscript, provided

as “Supplemental Material not for review,” includes an analysis of the cellular composition ratio using FACS (data presented right). Additionally, the study reports that the respiratory organoids exhibit immature cilia structures due to their fetal characteristics. TEM analysis confirmed that the cilia are shorter in length.

[Figure removed by editorial staff per authors' request]

Comment 16. Discussion. Line 245. To date mAbs have not worked as a therapeutic drug for RSV bronchiolitis in infants. Suggest rephrasing or include the human data that demonstrate lack of efficacy.

Response: In response to the reviewer's comment, we acknowledge that monoclonal antibodies are not effective as a treatment for RSV. Accordingly, we have further revised the manuscript to reflect this fact.

Comment 17. The authors should consider mentioning other airway organoid models used to study RSV and other respiratory viruses that are not based on iPS; perhaps a few sentences comparing and contrasting.

Response: In response to the reviewer's comment, we consider that organoids derived from both somatic stem cells and iPS cells are valuable tools for RSV research. Given the fetal characteristics of iPS cell-derived organoids, somatic stem cell-derived organoids are preferable when the use of adult cells is essential. However, unlike somatic stem cell-derived organoids, iPS cell-derived respiratory organoids contain mesoderm cell-derived fibroblasts, macrophages, and endothelial cells, allowing for the analysis of complex intercellular networks. Additional discussion on this aspect has been included in the revised manuscript. Please see lines 230-247.

Reviewer #2

Comment 1. In the results sections the authors need to provide a lead-up which is justification and the background prior to outlining the results. This gives the work context. For example the CRISPR KO of MDA5 is interesting work but they provide no justification for why they do the experiments. They claim that another study, briefly noted at the end of the section, found that RIG-I was ore antiviral than MDA5 but they do no experiments examining RIG-I so there is no comparator.

Response: Thank you for your comment. As this study focuses on the evaluation of anti-RSV F antibodies, the MDA5-related data has been removed. Given space limitations and the inability to provide a detailed background on MDA5, these data will be presented in a separate publication. In this study, our primary focus remains on the comparative analysis of anti-RSV F antibodies using respiratory organoids.

Comment 2. In line 187 they state that the RSV Mabs are therapeutic. They are not, they are only licensed for prevention of RSV, they are prophylaxis. This fact needs to be reflected in the introduction as well where they introduce remdesivir. They don't need to introduce remdesivir because it is a completely different type of drug and doesn't work against RSV.

Response: We agree with the reviewer's comments. Because anti-RSV F antibodies are preventive rather than therapeutic agents, we have corrected the relevant sections of the manuscript. Additionally, information regarding remdesivir has been removed. Please refer to the revised introduction section for these updates.

Comment 3. The abstract needs significant work. For example the first sentence is too awkward and needs to be rewritten. The results must be referred to in the past tense, they are currently referred to in the present tense. They do not encompass all of the results shown in the paper in the abstract the way it is written. The abstract makes the paper seem as though it is a much smaller study than it is. They say that they did ScSEQ to find the infected cells but this method also was used to characterize the model system. In summary, The abstract is difficult to understand, it doesn't adequately cover the contents of the paper, and needs to be rewritten.

Response: We completely rewrote the abstract.

Comment 4. In the figures, what do the authors mean by RSV-FP? Do they mean this is RSV-F and RSV-P or just RSV-F as is the norm for this protein. If it is just RSV-F then merely write 'RSV-F.' If it is P protein as well include a comma or such to distinguish the two proteins.

Response: We apologize for any confusion caused by the notation in the figures. In the revised manuscript, the abbreviation “P” for protein has been removed to ensure clarity.

Comment 5. JESS analysis in the materials and methods section needs to have a new subtitle. It should be called Western blot or Capillary based protein analysis. Something in this regard. JESS is the name of the system and not the method. The materials and methods needs more references as does the entire paper. They include catalogue numbers which is good but they don't cite the literature that first describes the method.

Response: The description of JESS has been revised, and additional relevant references have been included in the Methods section.

Comment 6. In the discussion or results the authors should compare the EC50s of the Mabs in their system compared to other systems. Are they the same, similar, more sensitive?

Response: Anti-RSV F antibodies were evaluated in both respiratory organoids and HEp-2 cells (see data below). When treated with the same antibody concentration, respiratory organoids exhibited a lower TCID50 value compared to HEp-2 cells. This suggests that respiratory organoids may provide a more sensitive platform for evaluating the effects of anti-RSV F antibodies than HEp-2 cells.

[Figure removed by editorial staff per authors' request]

Comment 7. In figure 1G what are some of the genes represented by the dots? Why not outline the highest and lowest expressed genes? Otherwise it is entirely moot as to why one would do the single cell SEQ in the first place.

Response: Following the reviewer's suggestion, gene names have been added to the volcano plot. Please see Figure 2C.

[Figure removed by editorial staff per authors' request]

Comment 8. In summary, this paper needs a very significant amount of work. All section needs to be rewritten, the data need be highlighted more and analyzed more in the results and discussed with reference to more studies in the discussion. This is not an exhaustive list.

Response: We have revised all sections of the manuscript, including the figures, and we hope these changes have improved the clarity and comprehensibility of the paper.

Reviewer #3

Comment 1. The title and abstract poorly reflect the content of the manuscript.

Response: The title and abstract have been thoroughly revised.

Comment 2. The authors discuss RSV-specific monoclonal antibodies as therapeutic agents, whereas these are licensed (and clinically effective) for prophylactic use (see manuscript line 77/82).

Response: We agree with your comment. Anti-RSV F antibodies are used as a prophylactic, and the revised manuscript no longer discusses their potential use as a therapeutic agent.

Comment 3. In many experiments the experimental design is unclear, the manuscript contains insufficient information to reproduce the experiments. The authors describe the differentiation of airway organoids from iPSCs using a matrigel-based method. It is not clear if the differentiation is performed in 2D or 3D. In addition, it is not clear which part of the respiratory tract these organoids should reflect, nasal, tracheal, bronchial or alveoli.

Response: We have provided a more detailed description of the method for generating respiratory organoids, as well as the protocol for infection experiments using respiratory organoids (Fig. 1A). As shown by scRNA-seq analysis, respiratory organoids consist of both airway epithelial cells and alveolar epithelial cells (Fig. S3A). A manuscript detailing the development of respiratory organoids is currently under peer review in another journal. This manuscript was provided as “Supplemental Material not for review.”

[Figures removed by editorial staff per authors' request]

Comment 4. If the major aim was to evaluate antiviral agents for RSV in respiratory organoids, it is unclear why the authors chose RNAseq as primary read-out. It would have been much more cost-effective to use immunofluorescence staining of infected cells.

Response: To assess the efficacy of anti-RSV F antibodies, a TCID50 assay was conducted, as detailed in the revised manuscript. Please see Figure 3B.

[Figure removed by editorial staff per authors' request]

Comment 5. The introduction contains insufficient background information. Information on

the background of RSV is completely lacking. The text focuses on antivirals and preventive monoclonal antibodies, until (in line 103) the authors mention for the first time that they analyzed the host response to RSV infection using organoids. The purpose of this analysis is not connected to the aim of the study. It is not clear to me (and not explained by the authors) why iPSC-derived respiratory organoids contain macrophages.

Response: As you suggested, we have added background information on RSV and anti-RSV antibodies to the introduction section. Please see lines 67-72 and 92-98. In our previous study, we developed respiratory organoids containing macrophages; however, the data remain unpublished and are provided as “Supplemental Material not for review.” In the current study, we confirmed that RSV infection experiments can be conducted using our respiratory organoids and subsequently evaluated the efficacy of anti-RSV antibodies.

Comment 6. Results: line 109 is the first place where iPSCs are mentioned, and the first three results paragraphs focus completely on analysis of host responses. RSV subgroups are suddenly introduced in line 114, but were not mentioned in the introduction.

Response: The rationale for conducting the host response analysis is provided in the introduction section. Please see lines 73-87. Due to the difficulty in using RSV-A and RSV-B at the same MOI, only RSV-A was used in the revised manuscript.

Comment 7. Results: the legend to figure 1 mentions that samples for RNAseq were collected four days after RSV infection. The authors do not explain how this time point was chosen, and do not include a growth curve of RSV in their iPSC-derived respiratory organoids.

Response: Following RSV infection of respiratory organoids, we conducted TCID₅₀ assays over time. The TCID₅₀ values peaked at 3 to 4 dpi (Fig. 1B), which led us to determine that performing RNA-seq analysis at 4 dpi was most appropriate.

[Figure removed by editorial staff per authors' request]

Comment 8. In line 137 -157 the authors directly compare host responses to RSV A and B strains. Again, it is unclear how this relates to the objectives of the manuscript. Also, it is unclear if differences between two strains can be translated into differences between subgroups.

Response: Due to the difficulty in using RSV-A and RSV-B at the same MOI, only RSV-A was employed in the revised manuscript. Additionally, comparing RSV-A and RSV-B would require the use of multiple strains from each genotype. We believe that the

differences between RSV-A and RSV-B cannot be fully explained by the results presented in this study alone. Therefore, in the revised manuscript, we have decided not to compare RSV-A and RSV-B.

Comment 9. In lines 161-178 the authors try to elucidate the mechanism of RSV-specific innate response. Again, it is unclear how this relates to the objectives of the manuscript.

Response: The purpose of this study is to demonstrate that respiratory organoids can be used in RSV infection experiments and to conduct a comparative analysis of anti-RSV antibodies. Since analyzing the mechanism of the innate immune response during RSV infection falls outside the scope of this study, we have decided to publish it as a separate paper.

Comment 10. Results: in lines 196-237 the authors describe they performed RNAseq analysis of RSV infected organoids treated with nirsevimab or palivizumab. The aim of this experiment is completely unclear to me. As described by the authors (cited in my major comment 2), these monoclonal antibodies were developed for prophylactic use, not for therapeutic use.

Response: As the reviewer pointed out, the primary objective is to evaluate whether anti-RSV F antibodies can prevent RSV infection. In the revised manuscript, we conducted TCID₅₀ assays on respiratory organoids treated with anti-RSV F antibodies. Additionally, we performed scRNA-seq analysis to assess whether RSV mRNA was detected in the anti-RSV F antibody-treated respiratory organoids.

Comment 11. The discussion completely lacks comparison with the international literature, either in comparison with other organoid models for RSV, other RNAseq studies for RSV or other model systems used for evaluation of antivirals or antibodies to RSV. Instead, the authors have included references of the literature in their results section. I would strongly recommend to keep results and discussion separate. The authors should discuss their model systems with models for culture of differentiated epithelial cells at air-liquid interface.

Response: We have included a discussion of other organoid models in the revised manuscript. In addition to organoids, we also addressed air-liquid interface models and discussed the differences between our model and others. Please see lines 230-247.

Specific comments

Comment 12. Title: remove 'therapeutic' and add 'iPSC-derived'

Response: The title has been corrected. New title: Human induced pluripotent stem cell-derived respiratory organoids as a model for respiratory syncytial virus infection.

Comment 13. Running title: remove 'lung' (no lung tissue was used)

Response: We fixed running title. New running title: Modeling RSV infection with respiratory organoids.

Comment 14. Abstract: remove EC90 data.

Response: In the revised manuscript, the EC90 value was removed from the abstract.

Comment 15. In line 53 instead of 'them' rather write RSV

Response: We have revised the manuscript as instructed. Please see line 55.

Comment 16. Type in line 80, change nirsavimab to nirsevimab

Response: We have revised the manuscript as instructed. Please see line 96.

Comment 17. Figure 1C: it is unclear to me what is stained by the antibody to acetylated-alpha-tubulin, but it does not look like cilia. It would be useful to match the RNA-seq data with convincing immunofluorescence staining of the different cell types in the organoid cultures.

Response: As noted by the reviewer, the morphology of acetylated α -tubulin-positive cells is immature, likely due to the fetal nature of the iPSC-derived respiratory organoids. Characterization of these organoids is being detailed in a manuscript currently under submission to another journal. The gene expression profile of the iPSC-derived respiratory organoids closely resemble that of fetal lungs (Right figure). In that manuscript, FACS analysis was also performed to validate the cellular composition analysis obtained through scRNA-seq. This manuscript was provided as "Supplemental Material not for review."

[Figure removed by editorial staff per authors' request]

Comment 18. Figure 1D: I do not understand the added value of the TEM data. The circular particle shape is concerning, as RSV particles are considered to be filamentous in vivo.

Response: The images in Figure 1D represent transversely sectioned filamentous virions, although they looked spherical.

Comment 19. It is not specified why the N protein for RSV-A is measured, but the P protein for RSV-B for figure S3.

Response: In the revised manuscript, we chose not to conduct a comparative analysis between RSV-A and RSV-B. Instead, we performed immunohistochemistry and JESS analysis using anti-RSV F antibodies. Please see figure 1D.

[Figure removed by editorial staff per authors' request]

In the revised manuscript, we decided not to conduct a comparative analysis between RSV-A and RSV-B. In the revised manuscript, we performed immunohistochemistry and JESS analysis using anti-RSV F antibodies.

Comment 20. Lines 193-196 (and tables): the authors should reconsider the precision of their results. I find it unrealistic to report these with two digits behind the comma.

Response: The description of the results concerning anti-RSV F antibodies has been revised. Please see lines 153-159.

Comment 21. Line 369: specify passage number. Is this virus adapted to in vitro cell culture?

Response: The RSV used in this study was passaged only a few times in HEp-2 cells and has not been adapted to *in vitro* cell culture.

Comment 22. The authors describe there is room for improvement in the development of antiviral drugs. However, no new compounds are tested and the existing ones have already been tested on airway organoids/HAE cells.

Response: As you pointed out, no new compounds were evaluated in this study. The manuscript has been revised accordingly.

Comment 23. The authors claim their model to be useful for testing therapeutic agents against RSV. However, a big downside of organoid models is that they are costly and time-consuming. The paper would be stronger if the authors describe why this model is so much better over the now often used immortalized cell lines, and how this model can be used in high-throughput to be able to test all these different therapeutic agents. Comment 1. In most cases, epithelial cells are the primary target of respiratory viruses, including human coronaviruses the authors have investigated here in the study. Fibroblasts, blood vessels and immune cells are involved due to their crosstalk with epithelial cells. during in vivo infections, these cell populations may not have direct exposure to the viruses. Not clear on the

procedure of virus inoculation in the respiratory tissue system, I assume that the organoids were incubated with virus solution, which means the virus inoculation was very different from real-life scenarios of human airway exposure to viruses. Based on our experience, fibroblasts, which are not supposed to be infected in vivo, are very permissive to viruses, including coronaviruses. This issue has to be clarified in the first place.

Response: The advantages and disadvantages of organoids compared to cell lines have been newly described in the introduction section. While organoids offer superior accuracy in analyzing host responses, they are less efficient than cell lines in terms of throughput. Characterization of the human iPSC-derived respiratory organoids used in this study is being detailed in a manuscript currently under submission to another journal. Respiratory epithelial cells are located on the apical side, while fibroblasts reside in the lumen of the organoid. Thus, it is likely that fibroblasts are rarely directly exposed to the virus.

March 5, 2025

Re: Life Science Alliance manuscript #LSA-2024-02837-TR

Dr. Kazuo Takayama
Kyoto University
Shogoin Kawaharacho 53, Sakyo-ku
Kyoto 6068507
Japan

Dear Dr. Takayama,

Thank you for submitting your revised manuscript entitled "Human iPS cell-derived respiratory organoids as a model for respiratory syncytial virus infection" to Life Science Alliance. The manuscript has been seen by the original reviewers whose comments are appended below. While the reviewers continue to be overall positive about the work in terms of its suitability for Life Science Alliance, some important issues remain.

Our general policy is that papers are considered through only one revision cycle; however, we are open to one additional short round of revision. Please note that I will expect to make a final decision without additional reviewer input upon re-submission. Reviewer 3's point #1 does not need to be addressed.

Please submit the final revision within one month, along with a letter that includes a point by point response to the remaining reviewer comments.

To upload the revised version of your manuscript, please log in to your account: <https://lsa.msubmit.net/cgi-bin/main.plex>
You will be guided to complete the submission of your revised manuscript and to fill in all necessary information.

B. MANUSCRIPT ORGANIZATION AND FORMATTING:

Sincerely,

Reviewer #1 (Comments to the Authors (Required)):

The authors responded appropriately to the comments and revised the manuscript accordingly. I have no additional comments.

Reviewer #2 (Comments to the Authors (Required)):

The first word in the introduction the authors say "RSV infections, like the coronavirus disease 2019 (COVID-19) and 69 influenza virus infection (flu)..." Rather it would make sense if they wrote "Respiratory infections, like the corona..." Otherwise I am happy with the manuscript

Reviewer #3 (Comments to the Authors (Required)):

The revised version of the manuscript by Hashimoto and colleagues has been much improved over the original version. Most of the responses to the comments of the reviewers are to the point and are accompanied by revisions in the manuscript text. However, some responses are incomplete, and do not fully address the review comment. In my opinion the manuscript still needs serious revision before it is suitable for publication.

1. The related manuscript that was sent with this revision does provide more background to the current manuscript. Therefore, I believe that the current manuscript should not be resubmitted until the other is accepted for publication. The authors should integrate that study in the introduction and explain what was already shown for the model. This should also include the observations made in their RSV infection studies.
2. The authors should re-assess the central message of the current manuscript: is it aimed at describing a novel model for studying RSV pathogenesis or describing a model for screening novel antiviral compounds? In my opinion the second is hard to sell, and I would advise the authors to choose the first option. The authors should end the introduction with a short paragraph of the objectives of the manuscript, that should form a clear bridge to the first paragraph of the results. This paragraph should acknowledge that they had already shown in the previous manuscript (here submitted as "related") that they were successful in performing RSV infections in the model. The authors can emphasize the productive virus infection shown here by re-isolation.
3. The authors have now added information on multiplicity of infection, it would help if they also specify the absolute viral inoculum (in TCID50 added per well).
4. The microscopic images in the manuscript are poor and should be revised. The structure of the organoids cannot be recognized from the images in the current figure 2A and B. It is unclear if the structures contain tight junctions, have stains with anti-ZO-1 been performed? Is it true that the infected cells are mostly inside the structures, not at the epithelium? (Fig2B)
5. The antibody sc-53029 (table S1) binds to alpha-tubulin, not acetylated alpha-tubulin. This likely explains the binding pattern, that is not restricted to cilia but also stains intra-cellular tubulin. The authors should select an alternative antibody (many are described in literature to specifically stain cilia), and repeat the staining.
6. The authors have added partial annotation to the genes with altered expression levels (new Fig 2C). They should add a supplementary table listing all differentially expressed genes, including the corresponding fold-change and adjusted p-values.
7. It is unexpected that neutralization by nirsevimab is inferior to that by palivizumab at 100ng/ml (Fig 3B). The authors should state clearly if this difference was reproduced in multiple experiments. In addition, it would be helpful to include one higher concentration (1ug/ml) to show if the infection can be neutralized completely.
8. A critical discussion of the current model with existing models (primary cells grown at air-liquid interface, airway organoids grown at air-liquid interface, apical-out organoids, etc) is still lacking. The authors focus on the claimed strengths of the model (especially the presence of immune cells), but fail to critically discuss the limitations. This would include the absence of an apical surface exposed to air, where mucus can accumulate (as in the respiratory tract of a living mammal). Moreover, a parenterally administered prophylactic monoclonal antibody will need to reach the respiratory epithelium from the submucosa of the airway, which cannot be mimicked in the current model.
9. The authors do mention that T and B cells are lacking, but the same is true for neutrophils, which have been described as crucial mediators of RSV-mediated inflammation. The authors should try to point out for which aims their model might be preferable over other existing models.

A point-by-point response to reviewer's comments

Reviewer #2

Comment 1. The first word in the introduction the authors say "RSV infections, like the coronavirus disease 2019 (COVID-19) and influenza virus infection (flu)..." Rather it would make sense if they wrote "Respiratory infections, like the corona..." Otherwise I am happy with the manuscript

Response: We sincerely appreciate the reviewer's valuable comments. In response, we have revised the manuscript accordingly. Please see line 67.

Reviewer #3

Comment 1. The related manuscript that was sent with this revision does provide more background to the current manuscript. Therefore, I believe that the current manuscript should not be resubmitted until the other is accepted for publication. The authors should integrate that study in the introduction and explain what was already shown for the model. This should also include the observations made in their RSV infection studies.

Response: Following the editor's instructions, we will not be responding to this comment. We appreciate the editor's thoughtful decision.

Comment 2. The authors should re-assess the central message of the current manuscript: is it aimed at describing a novel model for studying RSV pathogenesis or describing a model for screening novel antiviral compounds? In my opinion the second is hard to sell, and I would advise the authors to choose the first option. The authors should end the introduction with a short paragraph of the objectives of the manuscript, that should form a clear bridge to the first paragraph of the results. This paragraph should acknowledge that they had already shown in the previous manuscript (here submitted as "related") that they were successful in performing RSV infections in the model. The authors can emphasize the productive virus infection shown here by re-isolation.

Response: This study demonstrates the utility of our respiratory organoids in investigating RSV pathogenesis. In accordance with the reviewer's suggestion, we have revised the final paragraph of the introduction. Please see lines 101-104.

Comment 3. The authors have now added information on multiplicity of infection, it would help if they also specify the absolute viral inoculum (in TCID50 added per well).

Response: In the revised manuscript, the absolute viral inoculum has been specified. Please see lines 641, 659, 677.

Comment 4. The microscopic images in the manuscript are poor and should be revised. The structure of the organoids cannot be recognized from the images in the current figure 2A and B. It is unclear if the structures contain tight junctions, have stains with anti-ZO-1 been performed? Is it true that the infected cells are mostly inside the structures, not at the epithelium? (Fig2B)

Response: Additionally, we performed ZO-1 staining, which indicates the presence of tight junctions between epithelial cells (Right figure). We consider that RSV initially infects respiratory epithelial cells on the outer surface of the organoid. However, by 4 dpi, the epithelial layer is disrupted, likely allowing infected cells to be present within the lumen. Given our interest in the spread of RSV infection in respiratory organoids, we plan to investigate this further in our next study.

[Figure removed by editorial staff per authors' request]

Comment 5. The antibody sc-53029 (table S1) binds to alpha-tubulin, not acetylated alpha-tubulin. This likely explains the binding pattern, that is not restricted to cilia but also stains intra-cellular tubulin. The authors should select an alternative antibody (many are described in literature to specifically stain cilia), and repeat the staining.

Response: Thank you for your valuable comment. In the original study, we used an inappropriate antibody; therefore, in the revised manuscript, we have employed a new acetylated α -tubulin antibody. Using this updated antibody, we obtained the following staining image, confirming that the RSV F protein is expressed in acetylated α -tubulin-positive ciliated cells (Fig. 1F).

[Figure removed by editorial staff per authors' request]

Comment 6. The authors have added partial annotation to the genes with altered expression levels (new Fig 2C). They should add a supplementary table listing all differentially expressed genes, including the corresponding fold-change and adjusted p-values.

Response: As suggested by the reviewers, we have included fold-change values and

adjusted p-values in Table S2.

Comment 7. It is unexpected that neutralization by nirsevimab is inferior to that by palivizumab at 100ng/ml (Fig 3B). The authors should state clearly if this difference was reproduced in multiple experiments. In addition, it would be helpful to include one higher concentration (1ug/ml) to show if the infection can be neutralized completely.

Response: As suggested by the reviewer, we conducted additional experiments using higher antibody concentrations (1 µg/mL and 10 µg/mL) and found that these concentrations completely inhibited RSV infection (Fig. 3B).

Comment 8. A critical discussion of the current model with existing models (primary cells grown at air-liquid interface, airway organoids grown at air-liquid interface, apical-out organoids, etc) is still lacking. The authors focus on the claimed strengths of the model (especially the presence of immune cells), but fail to critically discuss the limitations. This would include the absence of an apical surface exposed to air, where mucus can accumulate (as in the respiratory tract of a living mammal). Moreover, a parenterally administered prophylactic monoclonal antibody will need to reach the respiratory epithelium from the submucosa of the airway, which cannot be mimicked in the current model.

Response: As the reviewer noted, our model lacks a sufficient mucus layer. Because the presence of a mucus layer significantly influences antibody dynamics, developing a new model that incorporates this feature is essential for accurately evaluating antibodies. We have further elaborated on this limitation in the discussion section. Please see lines 211-224.

Comment 9. The authors do mention that T and B cells are lacking, but the same is true for neutrophils, which have been described as crucial mediators of RSV-mediated inflammation. The authors should try to point out for which aims their model might be preferable over other existing models.

Response: As the reviewer noted, our model contains very few neutrophils. To accurately replicate RSV pathophysiology, a model incorporating neutrophils will be necessary. However, we believe that the inclusion of diverse respiratory epithelial cells and alveolar macrophages remains a key strength of our model. This point has been further discussed in the discussion section. Please see lines 230-234.

March 31, 2025

RE: Life Science Alliance Manuscript #LSA-2024-02837-TRR

Dr. Kazuo Takayama
Kyoto University
Shogoin Kawaharacho 53, Sakyo-ku
Kyoto 6068507
Japan

Dear Dr. Takayama,

Thank you for submitting your revised manuscript entitled "Human iPS cell-derived respiratory organoids as a model for respiratory syncytial virus infection". We would be happy to publish your paper in Life Science Alliance pending final revisions necessary to meet our formatting guidelines.

- please be sure that the authorship listing and order is correct
- please upload all figure files as individual ones, including the supplementary figure files; all figure legends should only appear in the main manuscript file
- please add the Twitter/X and Bluesky handles of your host institute/organization as well as your own or/and one of the authors in our system
- please upload your Tables in editable .doc or excel format
- include the supplementary reference in the main reference section
- please add your main, supplementary figure, and table legends to the main manuscript text after the references section;
- please add callouts for Figure S2A-B and Table S2 to your main manuscript text

A. FINAL FILES:

B. MANUSCRIPT ORGANIZATION AND FORMATTING:

Sincerely,

April 10, 2025

RE: Life Science Alliance Manuscript #LSA-2024-02837-TRRR

Dr. Kazuo Takayama
Kyoto University
Shogoin Kawaharacho 53, Sakyo-ku
Kyoto 6068507
Japan

Dear Dr. Takayama,

Thank you for submitting your Methods entitled "Human iPS cell-derived respiratory organoids as a model for respiratory syncytial virus infection". It is a pleasure to let you know that your manuscript is now accepted for publication in Life Science Alliance. Congratulations on this interesting work.

DISTRIBUTION OF MATERIALS:

Again, congratulations on a very nice paper. I have recently taken over from Eric Sawey as Executive Editor, and I was impressed with this interesting paper. I hope you found the review process to be constructive and are pleased with how the manuscript was handled editorially. We look forward to future exciting submissions from your lab.

Sincerely,

Tim Fessenden
Life Science Alliance